# Calorie restriction alters the mechanisms of radiation-induced mouse thymic lymphomagenesis

**Takafumi Nakayama[1,2,3], Masaaki Sunaoshi[1], Yi Shang[1], Mizuki Takahashi[1,3], Takato Saito[3], Benjamin J. Blyth[1¤a], Yoshiko Amasaki[1], Kazuhiro Daino[1], Yoshiya Shimada[1¤b], Akira Tachibana[3], Shizuko Kakinuma[1]***

**1** Department of Radiation Effects Research, National Institute of Radiological Sciences, National Institutes for Quantum Science and Technology, Chiba, Japan, **2** Department of Tumor and Diagnostic Pathology, Atomic Bomb Disease Institute, Nagasaki University, Nagasaki, Japan, **3** Graduate School of Science and Engineering, Ibaraki University, Mito, Japan

¤a Current Address: Present address: Cancer Research Division, Peter MacCallum Cancer Centre, Melbourne, Victoria, Australia
¤b Current Address: Present address: Institute for Environmental Sciences, Aomori, Japan
* kakinuma.shizuko@qst.go.jp

**Data Availability Statement:** The microarray data are deposited in the Gene Expression Omnibus database (http://www.ncbi.nlm.nih.gov/geo/) under accession no. GSE180952.

## Abstract

Calorie restriction (CR) suppresses not only spontaneous but also chemical- and radiation-induced carcinogenesis. Our previous study revealed that the cancer-preventive effect of CR is tissue dependent and that CR does not effectively prevent the development of thymic lymphoma (TL). We investigated the association between CR and the genomic alterations of resulting TLs to clarify the underlying resistance mechanism. TLs were obtained from previous and new experiments, in which B6C3F1 mice were exposed to radiation at 1 week of age and fed with a CR or standard (non-CR) diet from 7 weeks throughout their lifetimes. All available TLs were used for analysis of genomic DNA. In contrast to the TLs of the non-CR group, those of the CR group displayed suppression of copy-neutral loss of heterozygosity (LOH) involving relevant tumor suppressor genes (*Cdkn2a*, *Ikzf1*, *Trp53*, *Pten*), an event regarded as cell division–associated. However, CR did not affect interstitial deletions of those genes, which were observed in both groups. In addition, CR affected the mechanism of *Ikzf1* inactivation in TLs: the non-CR group exhibited copy-neutral LOH with duplicated inactive alleles, whereas the CR group showed expression of dominant-negative isoforms accompanying a point mutation or an intragenic deletion. These results suggest that, even though CR reduces cell division–related genomic rearrangements by suppressing cell proliferation, tumors arise via diverse carcinogenic pathways including inactivation of tumor suppressors via interstitial deletions and other mutations. These findings provide a molecular basis for improved prevention strategies that overcome the CR resistance of lymphomagenesis.

**Funding:** The Ministry of Education, Culture, Sports, Science and Technology of Japan (https://www.kenkyu.jp/nuclear/index.html)(AT) and the Japan Society for the Promotion of Science (https://www.jsps.go.jp/english/index.html) (JP15H01834 and JP21H04932) (SK). The funders and sponsors did not play a role in the study design, data collection and analysis, decision to publish, or preparation of the manuscript.

**Competing interests:** The authors have declared that no competing interests exist.

## Introduction

Calorie restriction (CR) increases lifespan and suppresses age-related morbidities in various animals [1, 2]. CR has been reported to decrease not only spontaneous but also chemical- and radiation-induced carcinogenesis [3–5]. Several biological mechanisms for the extension of lifespan by CR have been reported. CR has been suggested to suppress the PI3K/Akt/mTOR signaling pathways involved in cell proliferation and protein synthesis [6, 7]. CR also represses age-related genomic and epigenomic instability related to decreased DNA damage repair activity [8, 9] and changes in DNA methylation status [10, 11]. The efficacy of CR is, nevertheless, not universal. Some tumors are CR resistant, such as rat adrenal medullary tumors [12] and mouse lymphomas [13]. Thus, understanding how to overcome the mechanism of resistance to the beneficial effects of CR will provide a promising strategy for disease prevention and extension of health expectancy.

Animal models are crucial for obtaining insights into the mechanism of CR resistance. Our previous study revealed that CR suppresses tumor development in mice that were exposed to a high dose of radiation in early life, even though those mice were put under CR as adults [14]. The protective effect of CR was prominent for tumors that generally develop later in life, *e.g.*, liver and lung tumors; in contrast, CR was not effective for suppressing tumors such as thymic and other lymphomas which arise relatively early after exposure [14]. Thus, mechanisms must exist by which radiation-induced lymphomas develop even in the presence of CR. Previous studies have revealed that several key tumor suppressor genes (*Cdkn2a*, *Ikzf1*, *Trp53*, *Bcl11b*, *Pten*) and a proto-oncogene (*Myc*) are altered in radiation-induced TLs of mice [15–20]. *Cdkn2a* and *Trp53* encode tumor suppressor proteins that regulate the cell cycle and other cellular processes [21–23], *Ikzf1* and *Bcl11b* encode zinc-finger family transcription factors that regulate proliferation and differentiation of lymphoid cells [24, 25], and *Pten* encodes a protein that inhibits the PI3K-AKT-mTOR signaling pathway that regulates cell proliferation and survival [26, 27]. These tumor suppressor genes are inactivated via loss of heterozygosity (LOH) in radiation-induced thymic lymphoma [15–18, 20]. *Myc* positively controls cell proliferation and is overexpressed via mechanisms such as trisomy in various tumors including radiation-induced thymic lymphomas of mice [19, 28, 29].

The present study aimed to clarify the effect of CR on genomic alteration and to identify the distinct genomic alterations in radiation-induced TLs related to the resistance of the tumors to the suppressive effect of CR. Toward this goal, we compared genomic alterations in radiation-induced TLs that developed in the presence or absence of CR, focusing on previously reported genetic events including LOH, DNA copy-number variations, and point mutations affecting relevant tumor-related genes. The identification of distinct features of genomic alterations related to the resistance to CR is expected to provide a basis for improving the efficacy of CR interventions.

## Materials and methods

### Thymic lymphomas

The thymic lymphomas (TLs) used were cryopreserved samples stored in the Japan-Storehouse of Animal Radiobiology Experiments (J-SHARE) archive [30]. Those samples were obtained for experiments that were published previously [14] and for experiments carried out in the present study using a similar protocol. The incidence of TLs in that previous study [14] was 20% (12/60) and 14.3% (9/63) in the non-CR and CR groups, respectively, after exposure to 3.8 Gy of X-rays. No TL was observed in the non-irradiated group (non-CR group; 0/60, CR kcal group; 0/60). In the additional experimental group, all mice were irradiated with X-rays,

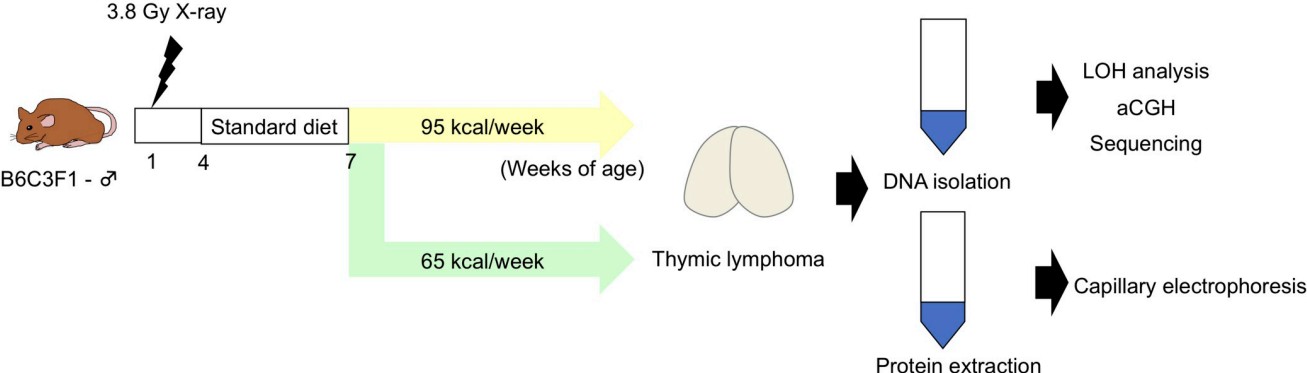

**Fig 1. Experimental design.** Male mice were irradiated with 3.8 Gy of X-rays at 1 week of age, weaned at 4 weeks of age, and fed a standard diet until 7 weeks of age. After 7 weeks of age, the mice were fed a 95 kcal/week or 65 kcal/week (calorie-restricted) diet. DNA copy number, loss of heterozygosity (LOH), point mutations, and protein expression of tumor suppressor genes in radiation-induced TLs were analyzed.

and the incidence of TL was 17.2% (17/99) and 14.1% (14/99) in the non-CR and CR groups, respectively; all experiments were approved by the Institutional Animal Care and Use Committee (approval no. 07–1080) of the National Institutes for Quantum Science and Technology and conducted in accordance with the relevant guidelines. Briefly, male B6C3F1 mice were produced by crossing male C3H/HeNCrlCrlj and female C57BL/6NCrlCrlj mice (Jackson Laboratory Japan, Inc., Yokohama, Japan). The mice were irradiated with 3.8 Gy of X-rays at 1 week of age, weaned at 4 weeks of age, and fed a standard diet (MB-1; Funabashi Farm Co., Ltd., Funabashi, Japan) until 7 weeks of age. Mice were then divided into two groups and fed special diets formulated to administer energy at either 95 or 65 kcal/week (designated 95 or 65 kcal group, respectively) for the remainder of life (Fig 1). The 95 kcal/week feeding was comparable to *ad libitum* feeding of the standard diet, whereas the 65 kcal/week feeding provided 32% less energy from carbohydrates; the remaining nutrients were kept comparable [4]. The animals were sacrificed when they showed signs of moribundity by exsanguination under isoflurane anesthesia, and thymic lymphomas (*n* = 27 and 21 for the 95 kcal and 65 kcal groups, respectively) were analyzed.

## DNA isolation and LOH analysis

Genomic DNA was extracted from thymic lymphoma cells using the Maxwell 16 Tissue DNA Purification kit (Promega, Madison, WI, USA). PCR was performed using TAKARA Taq polymerase (Takara Bio Inc., Otsu, Japan) and primers for microsatellite markers (S1 Fig and S1 Table). The PCR program consisted of 94°C for 3 min, cycles of denaturation at 94°C for 30 sec, annealing for 30 sec, and extension at 72°C for 15 sec, followed by 72°C for 5 min (see S1 Table for details). PCR products were analyzed by electrophoresis with 3% NuSieve 3:1 agarose gels (FMC, Rockland, MA, USA) or by capillary electrophoresis (QIAxcel Advanced System; Qiagen, Hilden, Germany).

## DNA copy-number analysis

Array-based comparative genomic hybridization (aCGH) was performed using customized 8 × 60 k microarrays (#046808; Agilent Technologies, Santa Clara, CA, USA), which contained probes that tiled across the whole genome (average density, 1 probe per 74 kb) and covered TL-relevant genes (*Cdkn2a*, *Ikzf1*, *Trp53*, *Bcl11b*, *Pten*) with high density (1 probe per 100–200

bp) as previously described [31, 32]. Fluorescence labeling of DNA, microarray hybridization, and microarray washing after hybridization were performed according to the manufacturer's protocol (Agilent Technologies, version 7.3). Microarrays were scanned using a microarray scanner (G2565BA, Agilent Technologies). Signal intensities were measured and evaluated using Feature Extraction software 10.5.1.1 (Agilent Technologies) and Genomic Workbench software 7.0.4.0 (Agilent Technologies).

## Reverse transcription PCR and sanger sequencing

Total RNA was extracted from thymic lymphoma cells using the Maxwell 16 RNA Purification kit and used for reverse transcription with Superscript III Reverse Transcriptase (Invitrogen, Carlsbad, CA, USA) and random hexamers. PCR was performed using primers for *Ikzf1*, *Trp53*, *Pten*, and *Gapdh* as described previously [16, 31], with expression of *Gapdh* used as an internal control. PCR products were electrophoresed through agarose gels containing ethidium bromide and photographed using a digital imager (Amersham Imager 600; GE Healthcare, Chicago, IL, USA). PCR products were directly sequenced using the Big Dye Terminator v3.1 kit (Applied Biosystems, Foster City, CA, USA) and 3500xL Dx Genetic Analyzer (Applied Biosystems). The primers used for PCR and sequencing analysis are shown in S2 Table.

## Capillary-based immunoassay

Expression of IKZF1, TRP53, and PTEN was analyzed by capillary-based western assays (WES System; ProteinSimple Inc., San Jose, CA, USA) according to the manufacturer's instruction using a 12–230 kDa Separation Module, $8 \times 25$ capillary cartridges (ProteinSimple) and either the Anti-Rabbit or Anti-Goat Detection Module (ProteinSimple). In brief, the protein content of TL cell extracts was quantified by the BCA Protein Assay kit (Pierce Inc., Rockford, IL, USA). Extracts were diluted to 200 µg/mL and heated at 95˚C for 5 min. Capillary electrophoresis of cell extracts, blocking reagent, primary antibodies, peroxidase-conjugated secondary antibodies, and chemiluminescent substrate were sequentially applied using the instrument default settings. The following primary antibodies were used: anti-β-actin (diluted 1:25, #4967S, Cell Signaling Technology, Danvers, MA, USA), anti-IKZF1 (1:80, ab26083, Abcam, Cambridge, UK and 1:25, #5443, Cell Signaling Technology), anti-TRP53 (1:100, AF1355-SP, R&D Systems, Minneapolis, MN, USA), and anti-PTEN (1:250, #9552, Cell Signaling Technology). Secondary antibodies were anti-rabbit (DM-001, ProteinSimple) and anti-goat (1:100, #sc-2033, Santa Cruz Biotechnology, Dallas, TX, USA). Electrograms were analyzed and quantified using Compass software version 3.1.7 (ProteinSimple).

## Immunohistochemistry

Paraffin-embedded tissue sections (4-µm thick) were dewaxed in xylene and rehydrated through an ethanol series and subjected to antigen retrieval in 10 mM sodium citrate buffer (pH 6.0) by heating at 110˚C for 15 min using a laboratory pressure vessel (Decloaking Chamber NxGen, Biocare Inc., Pacheco, CA, USA). Tissues were incubated with a rabbit monoclonal antibody against Ki-67 (1:200; clone SP6, M3060, Spring Bioscience, Pleasanton, CA, USA) at 4˚C overnight and a peroxidase-conjugated secondary antibody (Histofine Simple Stain MAX PO Rabbit kit; Nichirei Biosciences, Tokyo, Japan) at room temperature for 30 min. Peroxidase activity was visualized by 3,3′-diaminobenzidine staining (Simple Stain DAB Solution, Nichirei Biosciences), and tissues were counterstained with hematoxylin.

## Evaluation of immunohistochemical staining

After immunohistochemistry, the slides were digitalized using the NanoZoomer-XR slide scanner (Hamamatsu Photonics, Hamamatsu, Japan) and stored in the J-SHARE archive [30] in NDPI format. Each whole-slide image was divided into smaller tiles (regions of interest, ROI) that covered the entire image, and single images representing individual cases were captured independently at 40× magnification in JPEG format using NDP.view2 software (Hamamatsu Photonics). Ki-67 protein was analyzed in two steps. First, some captured ROI were evaluated visually by two researchers (T.S. and M.S.). Semiquantitative, 4-grade scoring was applied, ranging from negative (0) to strong (3+) immunoreactivity of the thymocytes. Second, the percentage of Ki-67$^+$ thymocytes was determined by automated scoring with Tissue Studio version 3.6.1 software (Definiens, Munich, Germany). Therein, staining thresholds (hematoxylin, 0.07; DAB density, 0.164; medium/low, 0.29; high/medium, 0.45) and morphological filtering (elliptic shape, <0.3; exclusion area, <3; typical nuclear size, 16) were set so that thymocytes within the ROI were recognized in a manner similar to the first analysis. The weight of Ki-67$^+$ cells in a thymus was calculated by multiplying the weight of the individual thymus by the percentage of Ki-67$^+$ cells in a slide.

## Statistical analysis

Frequency of LOH, DNA copy-number aberrations, and mutations between the 95 and 65 kcal groups were evaluated by two-sided Fisher's exact test using R software [33] with the graphical user interface EZR (Saitama Medical Center, Jichi University, Saitama, Japan). The results were considered significant at $p < 0.05$.

# Results

## CR reduces copy-neutral LOH and retains interstitial deletions in radiation-induced TLs

To obtain insights into the mechanism of resistance to CR-induced tumor suppression, genomic alterations involving relevant tumor suppressor genes (*Cdkn2a*, *Ikzf1*, *Bcl11b*, *Pten*) were analyzed in radiation-induced TLs that developed under non-CR (95 kcal) and CR (65 kcal) conditions by combining PCR-based LOH analysis and aCGH analysis. This approach successfully identified a number of terminal deletions, interstitial deletions, and copy-neutral LOHs as well as intragenic deletions (Fig 2A and 2B, Table 1, and S2 Fig) as detailed below.

*Cdkn2a*, which is located on chromosome 4, was affected by interstitial deletions in 15% and 24% and by copy-neutral LOHs in 19% and 0% of TLs in the 95 and 65 kcal groups, respectively; no terminal deletions were detected, and the total LOH frequency was 33% and 24%, respectively (Table 1), indicating a marginally significant CR-related decrease in the frequency of copy-neutral LOHs ($p = 0.059$). *Ikzf1*, located on chromosome 11, was affected by interstitial deletions in 41% and 52%, whereas it was associated with copy-neutral LOHs in 22% and 5% of TLs in the 95 and 65 kcal groups, respectively; a TL in the 95 kcal group (TL40) had a trisomy, an interstitial deletion, and a copy-neutral LOH (Fig 2A and 2B); no terminal deletions were detected, and the total LOH frequency was 59% and 57%, respectively (Table 1), reproducing the above tendency of CR-related decrease in copy-neutral LOHs ($p = 0.12$). *Bcl11b*, on chromosome 12, had interstitial deletions in 15% and 14%, copy-neutral LOHs in 22% and 33%, and any LOH in 82% and 91% of TLs in the 95 and 65 kcal groups, respectively (Table 1), indicating a negligible influence of CR. Finally, *Pten*, located on chromosome 19, was associated with interstitial deletions in 11% and 33%, copy-neutral LOHs in 7% and 5%, and any LOH in 19% and 38% of TLs in the 95 and 65 kcal groups, respectively

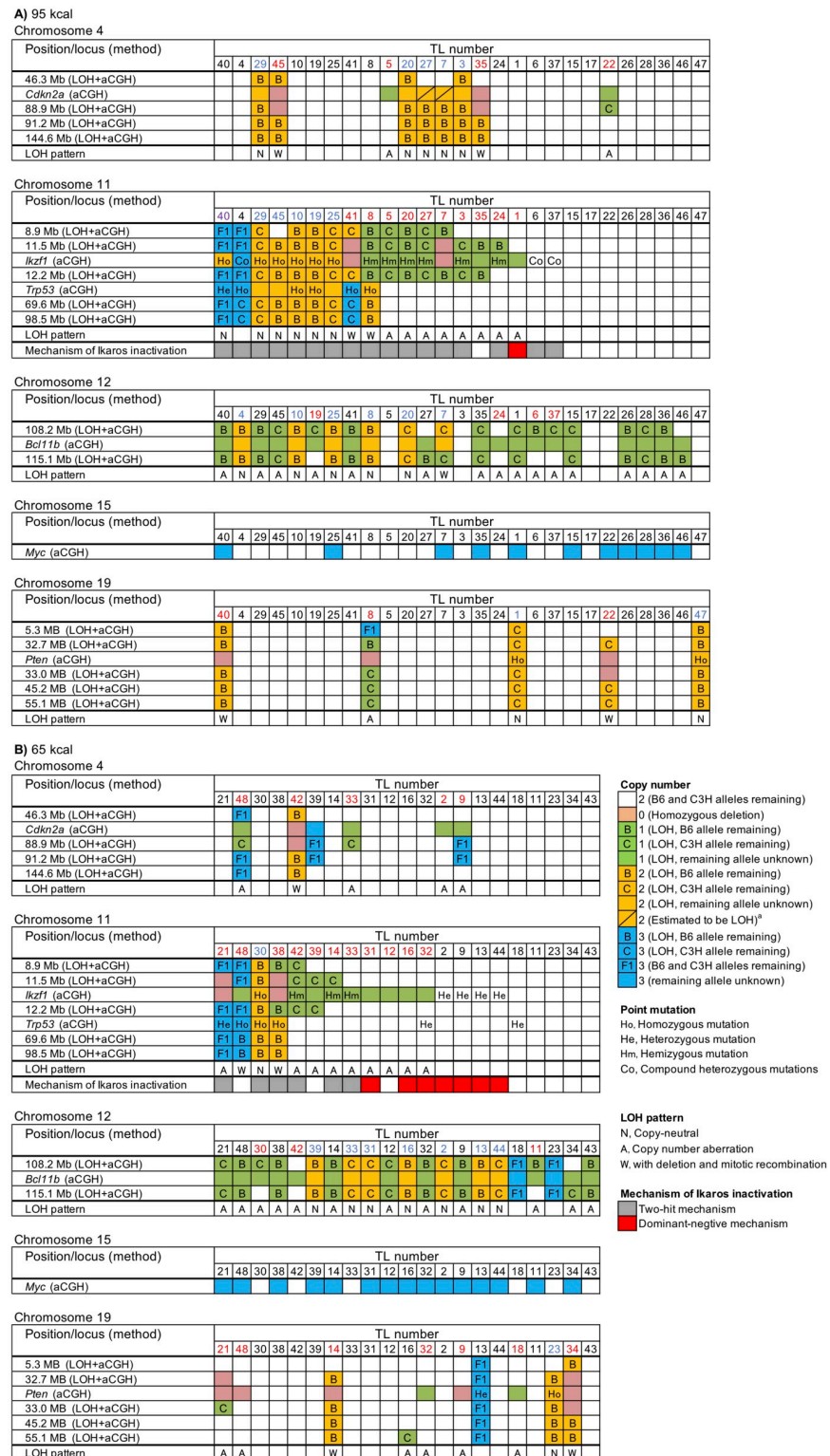

**Fig 2. Detailed results of DNA copy number, LOH, and point mutations in radiation-induced TLs.** Data are presented for chromosomes 4, 11, 12, 15, and 19 from 48 TLs (95 kcal, *n* = 27, (**A**); 65 kcal *n* = 21, (**B**)). Columns represent individual tumors, which are common in five panels for the same calorie group, ordered manually so that similar alterations are horizontally grouped. Rows represent integrated LOH and/or DNA copy-number (aCGH) data. Chromosomal locations noted in the leftmost columns are from the NCBI37/mm9 positioning. TL numbers in red

indicate TLs with interstitial deletion and blue indicates TLs with copy-neutral LOH. The TL number in purple (TL40 in chromosome 11) indicates a trisomy, an interstitial deletion, and a copy-neutral LOH. The TLs that did not fall into the aforementioned categories are shown in black. [a] *Cdkn2a* locus (88.9 Mb) of TL7 and TL27 was judged to be LOH from the result of the nearest marker at 88.9 Mb even though the upstream marker at 46.3 Mb did not exhibit LOH.

(Table 1), indicating a weak association of interstitial deletions with CR ($p = 0.08$); only one terminal deletion was detected (Fig 2A and 2B). Interestingly, four TLs (TLs 14, 22, 34, and 40 in Fig 2A and 2B) showed nullizygosity of *Pten* with two homozygotic copies of nearby loci (S3A Fig), implying that an interstitial deletion first occurred on one chromosome followed by its duplication by mitotic recombination or mis-segregation. On the other hand, four other TLs (TLs 8, 9, 21, and 48) harbored nullizygosity of *Pten* flanked by hemizygous deletions (S3B Fig), implying compound deletions.

On those four chromosomes, the copy-neutral LOHs often involved the telomere (Table 1, Fig 2A and 2B), suggesting their occurrence by mitotic recombination and mis-segregation, which are events associated with cell division. In fact, the cell proliferation marker Ki-67 was estimated to be expressed in fewer thymic cells during the period of CR (S4 Fig). With *Bcl11b* set aside which has a negligible influence of CR, CR significantly decreased TLs with any of the tumor suppressors affected by such copy-neutral LOHs (*i.e.*, those involving telomeres) (11/27 and 2/21 in the 95 and 65 kcal groups, respectively; $p = 0.02$). In addition, many of the LOHs with copy-number variation were interstitial deletions (Fig 2A and 2B), and CR did not significantly influence the frequency of TLs in which a tumor suppressor was affected by interstitial deletions (13/27 and 15/21 in 95 and 65 kcal groups, respectively). These data suggested that CR affects the mechanism of LOH by reducing the frequency of copy-neutral LOHs involving the telomere.

## CR increases the frequency of TLs that harbor copy-number gains involving *Myc*

Small numbers of full and partial trisomies (*i.e.*, copy-number gains) were observed on chromosomes 4, 11, 12, and 19 (Fig 2A and 2B, Table 1, and S3 Table) and other chromosomes (S5 Fig). Trisomy of chromosomes 1 and 14, as well as deletion of *Notch1* on chromosome 2, were identified in multiple TLs, but the frequency of these TLs was not significantly affected by CR (S3 Table). Interestingly, the frequency of amplification of the *Myc* locus on chromosome 15 was significantly increased by CR (41% and 71% in the 95 and 65 kcal groups, $p = 0.045$; S3 Table).

## Mutations and abnormal expression related to *Ikzf1*, *Trp53*, and *Pten*

To delineate other mechanisms affecting the relevant tumor suppressors, point mutations and expression of *Ikzf1*, *Trp53*, and *Pten* were examined and integrated with results of the aforementioned aCGH analysis. Most of the genomic alterations detected in this study, *i.e.*, point mutations [16] and aberrant isoforms of Ikzf1 [32, 34, 35], accumulation of mutant Trp53 [36], deletion [20], and point mutations [31] in the phosphorylation site of Pten, have been reported to be associated with lymphomagenesis. In addition, some of the genomic alterations detected in this study have already been shown to interfere with protein function [37–40]. CR tended [14] to be associated with following changes related to *Ikzf1* (Table 2): (i) an increase in TLs with copy-number variations (especially intragenic deletions), (ii) a decrease in TLs with nucleotide changes including missense, nonsense, and frameshift mutations (which occurred simultaneously in three TLs as in Table 3), (iii) a marginally significant decrease in TLs with

**Table 1. Genomic rearrangements affecting relevant genes in radiation-induced TLs developing under different CR conditions.**

| | Cdkn2a | | Ikzf1 | | Bcl11b | | Pten | |
|---|---|---|---|---|---|---|---|---|
| | 95 kcal | 65 kcal | 95 kcal | 65 kcal | 95 kcal | 65 kcal | 95 kcal | 65 kcal |
| Copy number[a] | | | | | | | | |
| 0 | 2 (7) | 1 (5) | 2 (7) | 2 (10) | 0 (0) | 0 (0) | 3 (11) | 5 (24) |
| 1 | 2 (7) | 4 (19) | 8 (30) | 9 (43) | 16 (59) | 12 (57) | 0 (0) | 2 (10) |
| 3 | 0 (0) | 1 (5) | 1[b] (4) | 0 (0) | 0 (0) | 2 (10) | 0 (0) | 1 (5) |
| LOH | 9 (33) | 5 (24) | 16 (59) | 12 (57) | 22 (82) | 19 (91) | 5 (19) | 8 (38) |
| Interstitial deletion | 4 (15) | 5 (24) | 11[b] (41) | 11 (52) | 4 (15) | 3 (14) | 3 (11) | 7 (33) |
| Copy-neutral[c] | 5 (19) | 0 (0) | 6[b] (22) | 1 (5) | 6 (22) | 7 (33) | 2 (7) | 1 (5) |
| Telomere involved | 5 (19) | 0 (0) | 5 (19) | 1 (5) | 5 (19) | 7 (33) | 2 (7) | 1 (5) |
| Other[d] | 0 (0) | 0 (0) | 0 (0) | 0 (0) | 11 (41) | 9 (43) | 0 (0) | 0 (0) |

[a]Numbers in parentheses, % ($n$ = 27 for 95 kcal, $n$ = 21 for 65 kcal).

[b]One TL had a trisomy, an interstitial deletion and a copy number neutral LOH.

[c]LOH that was detected by PCR and devoid of copy number variations detected by aCGH.

[d]Includes whole-chromosome loss and terminal deletion.

G>A mutations at the CpG site (caused by spontaneous deamination of 5-methylcytosine; 26% and 5% in the 95 and 65 kcal groups $p$ = 0.064), (iv) a decrease in TLs exhibiting null protein expression associated with nonsense/frameshift mutations and/or deletions, and (v) an increase in TLs with mutated IKZF1. These changes implied some influence of CR on the mechanism of inactivating *Ikzf1*, and this possibility is addressed further below.

In contrast with *Ikzf1*, alterations of *Trp53* were only negligibly affected by CR (Table 2). Nucleotide changes related to *Trp53* were observed at similar frequencies between the 95 and 65 kcal groups, which were mostly missense mutations including those of a hot-spot codon 245 (Table 3) [41] and other point mutations affecting the DNA-binding domain; TRP53, which is usually degraded to maintain low expression levels, accumulated in TLs harboring *Trp53* mutations (Table 3), which were observed again at similar frequencies between the two groups. CR tended to be associated with following changes related to *Pten*, including (i) a small increase in TLs with copy-number variations (mostly intragenic deletions), (ii) a negligible effect on TLs with nucleotide changes, which was small in number but involved the phosphatase domain, and (iii) a slight increase in TLs with protein abnormalities, implying an association between intragenic deletions and protein abnormalities (Tables 2 and 3).

## Classification of *Ikzf1* inactivation mechanisms by genomic alterations

The aforementioned analysis suggested that *Ikzf1*-inactivating mechanisms may participate in the resistance of CR-mediated tumor suppression. As it was unclear whether these alterations functionally inactivated IKZF1, the mechanisms of *Ikzf1* inactivation of individual TLs were re-evaluated and classified into the following three categories. The first category ('two-hit mechanism') met Knudson's two-hit hypothesis [42], in which one *Ikzf1* allele is lost first by a mutation and inactivation of the remaining allele follows. The second category ('dominant-negative mechanism') included mechanisms in which a mutant IKZF1 harboring a mutation in the N-terminal zinc-finger domain inactivates the intact IKZF1, encoded by the remaining *Ikzf1* allele, by forming a heterodimer. The third category ('unknown') included null expression of IKZF1 with unclear mechanisms. The results revealed that inactivation by the two-hit mechanism was significantly more prominent in the 95 kcal group than in the 65 kcal group (Fig 3). Herein, the dominant cases of point mutations combined with rearrangements and

**Table 2. TLs harboring mutations and protein abnormalities related to *Ikzf1*, *Trp53*, and *Pten*.**

|  | *Ikzf1* | | *Trp53* | | *Pten* | |
| --- | --- | --- | --- | --- | --- | --- |
|  | 95 kcal | 65 kcal | 95 kcal | 65 kcal | 95 kcal | 65 kcal |
| CNV[a] | 10 (37) | 11 (52) | 0 | 0 | 3 (11) | 7 (33) |
| Large deletion | 9 (33) | 6 (29) | 0 | 0 | 1 (4) | 3 (14) |
| Intragenic deletion | 1 (4) | 5 (24) | 0 | 0 | 2 (8) | 4 (19) |
| Nucleotide change | 15 (56) | 8 (38) | 6 (22) | 6 (29) | 2 (8) | 2 (10) |
| Missense mutation | 9[b] | 6 | 4 | 7[b] | 3[b] | 1 |
| Nonsense mutation | 5 | 0 | 0 | 0 | 0 | 0 |
| Frame shift | 6[b] | 2 | 0 | 0 | 0 | 0 |
| Splicing mutation | 0 | 0 | 2 | 0 | 0 | 1 |
| G>A at CpG | 7 (26) | 1 (5) | 0 | 0 | 0 | 1 (5) |
| Protein abnormalities | 18 (67) | 15 (71) | 6 (22) | 6 (29) | 6 (22) | 7 (33) |
| Null expression | 11 (41) | 5 (24) | - | - | 4 (15) | 6 (29) |
| Mutated protein | 7 (26) | 10 (48) | 6 (22) | 6 (29) | 2 (7) | 1 (5) |

[a]Copy number variations. Numbers in parentheses, % of TL ($n$ = 27 for 95 kcal, $n$ = 21 for 65 kcal).
[b]Multiple mutations in a single TL were counted individually.

rarer cases of multiple point mutations (such as TLs 4, 6, and 37; Table 3) were included. In contrast, the dominant-negative mechanism predominated in the 65 kcal group, which was significantly different from the 95 kcal group (Fig 3). Previously reported dominant-negative isoforms termed Ik6, Ik8, and Ik12 [16, 32, 35] were generated via intragenic deletions (TLs 1, 16, 31, and 32 in Table 3). The category 'unknown' was observed only in the 65 kcal group (Fig 3). Thus, *Ikzf1* was functionally inactivated in the majority of TLs analyzed, for which CR favored the occurrence of inactivation via the dominant-negative mechanism.

## Discussion

The current study explored the effect of CR on genomic alteration and the mechanism of resistance of radiation-induced TLs to CR-mediated tumor suppression by examining the genomic alterations in TLs that developed under non-CR and CR conditions. Radiation exposure occurred at 1 week of age, and CR was administered from 7 weeks of age. The genomic alterations included point mutations, copy-number gains, interstitial deletions, and copy-neutral LOHs. In particular, the frequency of copy-neutral LOHs involving *Cdkn2a*, *Ikzf1*, and *Pten* was reduced by CR, whereas interstitial deletions were not affected. Thus, at least part of the CR resistance was mediated by these interstitial deletions, which could bypass suppression by CR.

The results indicate that CR reduces tumorigenic mechanisms related to cell proliferation. First, CR suppressed the frequency of copy-neutral LOHs extending to the telomeric regions. These abnormalities are interpreted as mitotic recombination and chromosomal mis-segregation, which generally occur during cell division [43]. CR also weakly reduced the frequency of G>A mutations at the CpG sites of *Ikzf1*. This type of mutation generally occurs via deamination of 5-methylcytosine into thymine, which subsequently converts a C:G base pair into T:A upon DNA replication, and is thus methylation- and proliferation-dependent. CR indeed reduces cell proliferation in multiple organs of mice [44, 45], including thymus as confirmed herein (S4 Fig). In addition, CR is also reported to reduce DNA methylation [46]. Thus, suppression of cell proliferation is a plausible versatile mechanism by which CR inhibits multiple events generating relevant mutations.

**Table 3. Aberrations in *Ikzf1*, *Trp53* and *Pten*.**

| Group and TL No. | Ikzf1 CNV region | Ikzf1 RNA expression | Ikzf1 Nucleotide change | Ikzf1 Amino acid change | Ikzf1 Protein | Inact. Dominant negative | Inact. Two-hit | Inact. Unknown[a] | Trp53 CNV region | Trp53 Nucleotide change | Trp53 Amino acid change | Trp53 Protein | Pten CNV region | Pten Nucleotide change | Pten Amino acid change | Pten Protein |
|---|---|---|---|---|---|---|---|---|---|---|---|---|---|---|---|---|
| **95 kcal** | | | | | | | | | | | | | | | | |
| 1 | Ex 3–6 Del (He) | Ik6 | - | - | Ik6 | ○ | - | - | - | | | | | c.[764T>C(;) 799A>G] | p.[V255A (;) K267E] | Low |
| 3 | Large del (He)[b] | Low | c.664C>T[c] | p.R222X | Null | - | ○ | - | - | - | - | - | - | - | - | - |
| 4 | Amplification | - | c.[572A>G(;) 1048dupC] | p.[H191R(;) P350fs] | Mutant | - | ○ | - | - | c.649A>C | p.Y217D | Accumulation | - | - | - | - |
| 5 | Large del (He)[b] | - | c.637C>T[c] | p.R213X | Null | - | ○ | - | - | - | - | - | - | - | - | - |
| 6 | - | - | c.[979_989del(;) 989C>G(;) 1213_1214insGG(;) 1215C>G] | p.[L327fs(;) T330R(;) S405fs(;) S405R] | Null | - | ○ | - | - | - | - | - | - | - | - | Null[d] |
| 7 | Large del (HD)[b] | Null | - | - | Null | - | ○ | - | - | - | - | - | - | - | - | - |
| 8 | Large del (He) | Low | c.637C>T[c] | p.R213X | Null | - | ○ | - | - | c.733G>A | p.R245C | Accumulation | - | - | - | Null |
| 10 | - | - | c.1048dupC | p.P350fs | Null | - | ○ | - | - | c.773+2T>G | Splicing mutation | Accumulation | - | - | - | - |
| 15 | - | - | - | - | - | - | - | - | - | - | - | - | - | - | - | - |
| 17 | - | - | - | - | - | - | - | - | - | - | - | - | - | - | - | - |
| 19 | - | - | c.664C>T[c] | p.R222X | Null | - | ○ | - | - | c.637G>A | p.V213M | Accumulation | - | - | - | - |
| 20 | Large del (He)[b] | - | c.1389C>G | p.C463W | Null | - | ○ | - | - | - | - | - | - | - | - | - |
| 22 | - | - | - | - | Null | - | - | - | - | - | - | - | Ex2-9 Del (HD) | - | - | Null |
| 24 | Large del (He)[b] | - | c.1352_1353ins22bp | p.S451fs | Null | - | ○ | - | - | - | - | - | - | - | - | - |
| 25 | - | - | c.356G>A | p.C119Y | Mutant | - | ○ | - | - | - | - | - | - | - | - | - |
| 26 | - | - | - | - | - | - | - | - | - | - | - | - | - | - | - | - |
| 27 | Large del (He)[b] | - | c.664C>T[c] | p.R222X | Null | - | ○ | - | - | - | - | - | - | - | - | - |
| 28 | - | - | - | - | - | - | - | - | - | - | - | - | - | - | - | - |
| 29 | - | - | c.482T>C | p.L161P | Mutant | - | ○ | - | - | - | - | - | - | - | - | - |
| 35 | Large del (He)[b] | - | - | - | - | - | - | - | - | - | - | - | - | - | - | - |
| 36 | - | Low | - | - | N.D. | - | - | - | - | - | - | N.D. | - | - | - | N.D. |
| 37 | - | - | c.[428G>A[c](;) 1199_1200insGCCA] | p.[R143Q(;) A400fs] | Mutant | - | ○ | - | - | - | - | - | - | - | - | - |
| 40 | - | - | c.571C>T | p.H191Y | Mutant | - | ○ | - | - | c.508G>A | p.V170M | Accumulation | Ex 3-5Del (HD) | - | - | Null |
| 41 | Large del (HD)[b] | Null | - | - | Null | - | ○ | - | - | c.773+2T>C | Splicing mutation | Accumulation | - | - | - | - |
| 45 | - | - | c.550C>T[c] | p.R184W | Mutant | - | ○ | - | - | - | - | - | - | - | c.79T>A | p.Y27N | Low |
| 46 | - | - | - | - | - | - | - | - | - | - | - | - | - | - | - | - |
| 47 | - | - | - | - | - | - | - | - | - | - | - | - | - | - | - | - |
| Frequency of TLs with abnormalities[e] | | | | 15/27 (56) | 18/27 (67) | 1/27 (4) | 17/27 (63) | 0/27 (0) | | | 6/27 (22) | 6/27 (22) | | | 2/27 (7) | 6/27 (22) |
| **65 kcal** | | | | | | | | | | | | | | | | |

*(Continued)*

**Table 3.** (Continued)

| Group and TL No. | Ikzf1 CNV region | RNA expression | Nucleotide change | Amino acid change | Protein | Inactivation mechanism — Dominant negative | Two-hit | Unknown[a] | Trp53 CNV region | Nucleotide change | Amino acid change | Protein | Pten CNV region | Nucleotide change | Amino acid change | Protein |
|---|---|---|---|---|---|---|---|---|---|---|---|---|---|---|---|---|
| 2 | – | – | c.557A>C | p.D186A | Mutant | o | – | – | – | – | – | – | – | – | – | – |
| 9 | – | – | c.480_482del | p.L160del | Mutant | o | – | – | – | – | – | – | Ex 1-2Del (HD) | – | – | Null |
| 11 | – | – | – | – | – | – | – | – | – | – | – | – | – | – | – | – |
| 12 | Large del (He)[b] | – | – | – | – | – | – | – | – | – | – | – | – | – | – | – |
| 13 | – | – | c.485G>A[c] | p.R162Q | Mutant | o | – | – | – | – | – | – | – | c.210+1G>T | Splicing mutation | Null |
| 14 | Ex 1–2 Del (He) | – | c.51dupC | p.S18fs | Mutant | – | o | – | – | – | – | – | Ex 1 Del (HD) | – | – | Null |
| 16 | Ex 4 Del (He) | Ik12 | – | – | Ik12 | o | – | – | – | – | – | – | – | – | – | – |
| 18 | – | – | – | – | – | – | – | – | – | c.574A>T | p.I192F | Accumulation | Large del (He)[b] | – | – | Low |
| 21 | Large del (HD)[b] | Null | – | – | Null | – | o | – | – | c.385[A>G(;)791G>C] | p.[K129E(;)R264P] | Accumulation | Ex 1–2 Del (HD) | – | – | Null |
| 23 | – | Low | – | – | N.D. | – | – | – | – | – | – | N.D. | – | c.389G>A[c] | p.R130Q | N.D. |
| 30 | – | – | c.575T>C | p.L192P | Mutant | – | o | – | – | c.637G>A | p.V213M | Accumulation | – | – | – | – |
| 31 | Ex 3–5 Del (He) | Ik8 | – | – | Ik8 | o | – | – | – | – | – | – | – | – | – | – |
| 32 | Ex 3–6 Del (He) | Ik6 | – | – | Ik6 | o | – | – | – | c.698A>G | p.Y233C | Accumulation | Large del (He)[b] | – | – | Null[d] |
| 33 | Ex 7 Del (He) | – | c.557A>C | p.D186A | Mutant | – | o | – | – | – | – | – | – | – | – | – |
| 34 | – | – | – | – | Null | – | – | o | – | – | – | – | Large del (HD)[b] | – | – | Null |
| 38 | Large del (HD)[b] | Null | – | – | Null | – | o | – | – | c.637G>A | p.V213M | Accumulation | – | – | – | – |
| 39 | Large del (He)[b] | – | – | – | Null | – | – | o | – | – | – | – | – | – | – | – |
| 42 | Large del (He)[b] | Null | c.1270G>T | p.L424I | Null | o | – | – | – | – | – | – | – | – | – | – |
| 43 | – | – | – | – | – | – | – | – | – | – | – | – | – | – | – | – |
| 44 | – | – | c.575T>C | p.L192P | Mutant | – | o | – | – | – | – | – | – | – | – | – |
| 48 | Large del (He)[b] | – | – | – | – | – | – | – | – | c.404C>T | p.A135V | Accumulation | Ex 3–9 Del (HD) | – | – | Null |
| Frequency of TLs with abnormalities[e] | | | | 8/21 (38) | 15/21 (71) | 7/21 (33) | 6/21 (29) | 2/21 (10) | | | 6/21 (29) | 6/21 (29) | | | 2/21 (10) | 7/21 (33) |

Abbreviations: CNV, copy number variations; Ex, exon; Del, deletion; HD, homozygous deletion; He, heterozygous deletion; N.D., no data.

[a] Null IKZF1 expression via an unknown mechanism.

[b] Included in a large deleted region.

[c] Mutations at a CpG site.

[d] Null PTEN expression via an unknown mechanism.

[e] Numbers in parentheses, % of TL.

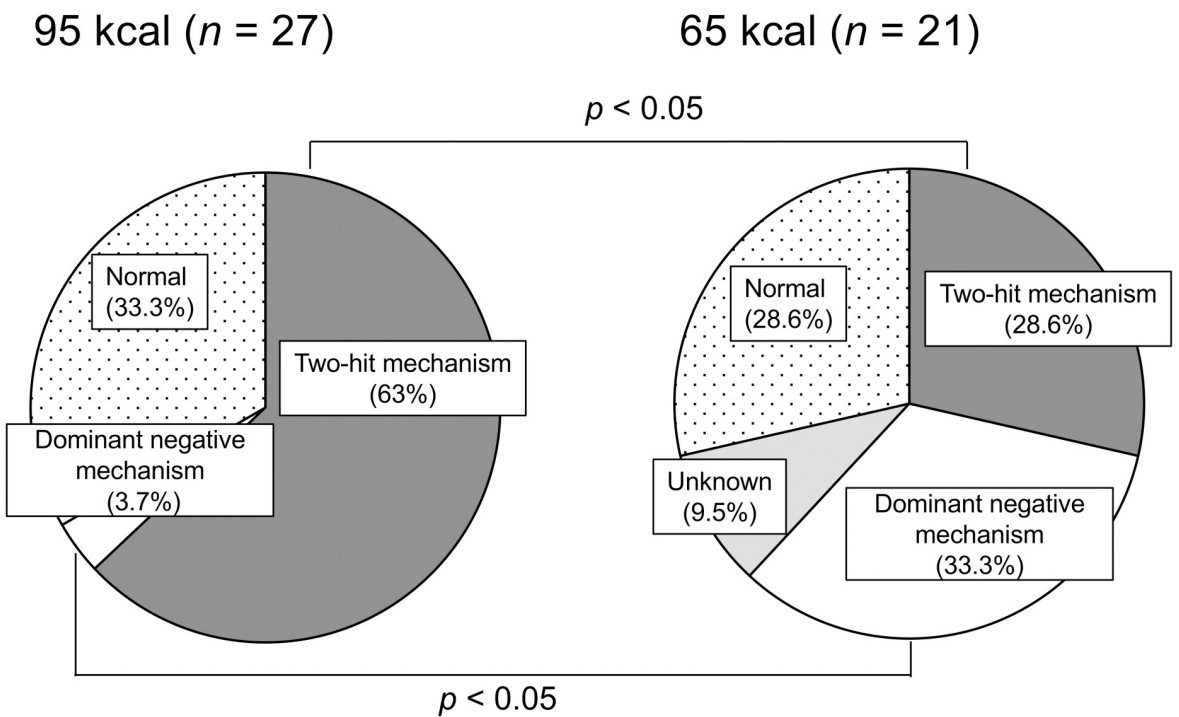

**Fig 3. Distribution of *Ikzf1* inactivation mechanisms determined from the results of LOH and sequence analyses.** *Unknown*, null expression of IKZF1 via an unknown mechanism.

Some genomic alterations herein were not affected by CR, providing evidence for the mechanism of CR resistance. The LOH on chromosome 12 affecting *Bcl11b* was most frequent among all alterations observed, and their patterns (*i.e.*, whether they were a copy-neutral LOH, terminal deletion, or interstitial deletion) did not depend on CR. LOH involving *Bcl11b* is reported to coincide with LOH of *Ikzf1* and *Pten* in radiation-induced TLs [31] and it is detectable within 30 days after irradiation, which is earlier than LOH of any other genes detected [19]. Therefore, LOH affecting *Bcl11b* is likely to have occurred before the start of CR at 7 weeks of age. Likewise, interstitial deletions on chromosomes 4, 11, and 19 also appeared regardless of the CR intervention. Interstitial deletion has been reported as a mutational signature of radiation exposure in multiple animal models including medulloblastoma of *Ptch1* heterozygous mice and kidney tumors of *Tsc2* heterozygous Eker rats [47–49]. In these models, the wild-type allele is often lost via copy-neutral LOH in sporadic tumors, whereas it is lost by interstitial deletions in tumors arising after radiation exposure [47–49] at a frequency that depends on the radiation dose [47]. Recently, papillary thyroid cancer after the Chernobyl nuclear accident was reported to exhibit an association between radiation dose and the length of genomic deletions [50]. Thus, the interstitial deletions observed herein were likely caused by X-ray exposure at 1 week of age and persisted in the tissue after initiation of CR at 7 weeks.

The present study indicates that functional alterations of *Ikzf1* constitute a major mechanism of lymphomagenesis, including the 'two-hit mechanism' and 'dominant-negative mechanism', which were characteristic of non-CR and CR conditions, respectively. Therein, the 'two-hit mechanism' involved mitotic recombination and mis-segregation, which was CR-sensitive as discussed above. The observation suggests that some copy-neutral LOHs formed after CR started. On the other hand, the 'dominant-negative mechanism' depended solely on point

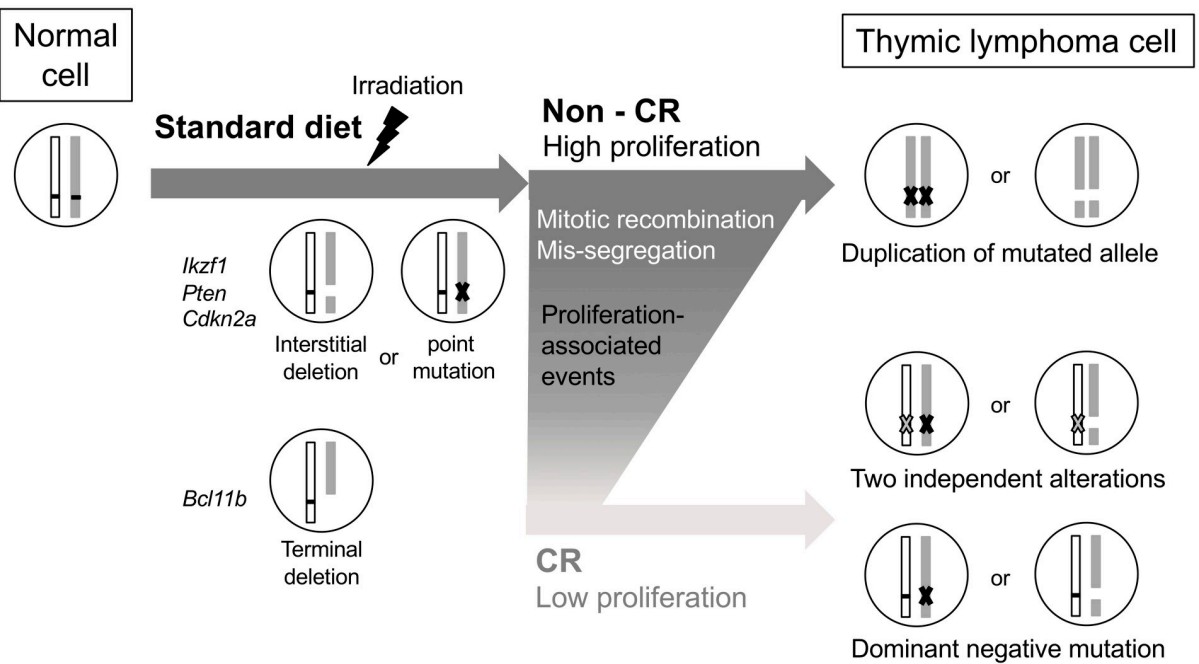

**Fig 4. Postulated mechanism of resistance of TL development to CR-mediated tumor suppression.** Interstitial deletions in tumor suppressor genes are induced by radiation; especially, the deletion of *Bcl11b* occurs before CR. Point mutations as well as mitotic recombination and chromosomal mis-segregation related to cell proliferation occur in the non-CR condition, which are reduced by CR. On the other hand, dominant-negative IKZF1 isoforms enable development of lymphoma cells under CR.

mutations and interstitial deletions, which were CR-insensitive. Thus, *Ikzf1* remains subject to inactivation by this alternative mechanism even if the 'two-hit mechanism' is suppressed by CR, enabling evasion from the CR-mediated reduction of tumorigenesis. The mechanism underlying the observed association between CR and copy-number gains of *Myc* is also of interest. Given that amplification of *Myc* is frequently observed in cancer cells [19] and increases proliferative activity [28, 29], it may play an important role in gaining extra proliferative activity under the suppressive CR condition. Mutations in other genes analyzed, including *Bcl11b*, *Cdkn2a*, and *Pten*, also upregulate cell proliferation [21, 25, 26]; herein, the mutations of *Cdkn2a* and *Pten* were mutually exclusive, supporting the cooperative nature of these genes.

We thus propose a scheme of thymic lymphomagenesis consisting of key genomic alteration events (Fig 4). First, an interstitial deletion of *Ikzf1*, *Pten*, or *Cdkn2a* occurs immediately after radiation exposure. LOH of *Bcl11b* then forms before initiation of CR, and active cell proliferation subsequently induces point mutations, mitotic recombination, and chromosomal mis-segregation. This activity is suppressed under CR, in which *Myc* amplification counteracts the CR-mediated anti-proliferative effect.

In conclusion, the present results suggest that the mutational mechanism of radiation-induced TL is affected by CR applied after radiation exposure. More specifically, CR suppresses genomic alterations caused via cell proliferation, whereas the genesis of radiation-induced interstitial deletions remains unaffected. In liver and lung, where cancer incidence is decreased by CR, suppression of proliferation-related genomic alterations may hinder the carcinogenic process. In the thymus, on the other hand, lymphomagenesis is promoted even under this effect of CR, with interstitial deletions and point mutations remaining undisturbed.

This molecular mechanism of resistance to CR-mediated tumor suppression will be useful for developing methodologies to control the risk of second cancer after radiotherapy of cancer patients by improving the tumor-preventive efficacy of CR.

## Supporting information

**S1 Table. Microsatellite markers used to analyze LOH.** Microsatellite marker positions are referenced to the NCBI37/mm9 mouse genome sequence.
(XLSX)

**S2 Table. Primers used in PCR and sequencing analyses.**
(XLSX)

**S3 Table. Number of TLs with trisomy.**
(XLSX)

**S1 Fig. Microsatellite markers used for the PCR-based LOH analysis.** Names and locations of the microsatellite markers are shown in black, and representative tumor suppressor genes are shown in red. Dots indicate kinetochores.
(DOCX)

**S2 Fig. An example of LOH that was detected by aCGH but not by PCR.** Left, diagram of chromosome 11, with the red bars indicating microsatellite markers to determine LOH. Right, DNA copy number analyzed by aCGH. The LOH was not detected by PCR because of its small size, residing between the nearest microsatellite markers.
(DOCX)

**S3 Fig. Examples of homozygous deletions of Pten.** Red bars, *Pten* locus. (A) A TL harboring a homozygous deletion formed via two different heterozygous deletions. (B) A TL harboring two copies of a deleted allele duplicated via mitotic recombination or mis-segregation.
(DOCX)

**S4 Fig. Cell proliferation in the thymus after CR was initiated.** (A) Representative images of Ki-67 immunostaining. Brown, immunopositive cells; blue, counterstaining (hematoxylin). Scale bars, 1 mm. (B) Changes in the mean percentage of Ki-67$^+$ cells in a tissue section over time. (C) Calculated weight of Ki-67$^+$ cells in a thymus. Error bars, standard error ($n$ = 3–6).
(DOCX)

**S5 Fig. Trisomy in TLs.** Chromosomes harboring trisomy in individual TLs are indicated by red boxes.
(DOCX)

## Acknowledgments

The authors thank the staff at the Laboratory Animal and Genome Sciences Section of the National Institutes for Quantum Science and Technology for animal management; Ms. Kanae Ogawa for the sampling part of thymic lymphoma; and Ms. Mutsumi Kaminishi, Yoshiko Amasaki, Mayumi Okabe, and Mayumi Shinagawa for their technical assistance. We thank Dr. Tatsuhiko Imaoka for his expertise and assistance throughout all aspects of our study.

## Author Contributions

**Data curation:** Takafumi Nakayama, Yi Shang, Mizuki Takahashi.

**Formal analysis:** Takafumi Nakayama, Masaaki Sunaoshi, Mizuki Takahashi, Takato Saito, Yoshiko Amasaki.

**Funding acquisition:** Akira Tachibana, Shizuko Kakinuma.

**Investigation:** Takafumi Nakayama, Masaaki Sunaoshi, Mizuki Takahashi, Takato Saito.

**Methodology:** Benjamin J. Blyth, Kazuhiro Daino.

**Project administration:** Shizuko Kakinuma.

**Resources:** Yi Shang.

**Supervision:** Benjamin J. Blyth, Kazuhiro Daino, Yoshiya Shimada, Akira Tachibana, Shizuko Kakinuma.

**Writing – original draft:** Takafumi Nakayama.

**Writing – review & editing:** Masaaki Sunaoshi, Shizuko Kakinuma.

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
