## [Decision Letter · Decision Letter 0]

23 Nov 2022

PONE-D-22-29050Calorie restriction alters the mechanisms of radiation-induced mouse thymic lymphomagenesisPLOS ONE

Dear Dr. Kakinuma,

Thank you for submitting your manuscript to PLOS ONE. After careful consideration, we feel that it has merit but does not fully meet PLOS ONE’s publication criteria as it currently stands. Therefore, we invite you to submit a revised version of the manuscript that addresses the points raised during the review process.

Manuscript has been reviewed by 3 experts in the field. All three have appreciated the questions asked in the manuscript and quality of data and conclusion from this work. At the same time, all three have raised some important concerns that I am fully agree and find them important for the wider appreciation of this work.    

We look forward to receiving your revised manuscript.

Kind regards,

Hari S. Misra

Academic Editor

PLOS ONE

Journal Requirements:

Additional Editor Comments:

Manuscript has been reviewed by 3 experts in the field. All three have appreciated the questions asked in the manuscript and quality of data and conclusion from this work. At the same time, all three have raised some important concerns that I fully endorse and are important for the better appreciation of this work.

Reviewers' comments:

Reviewer's Responses to Questions

**Comments to the Author**

1. Is the manuscript technically sound, and do the data support the conclusions?

Reviewer #1: Yes

Reviewer #2: Partly

Reviewer #3: Yes

2. Has the statistical analysis been performed appropriately and rigorously? 

Reviewer #1: Yes

Reviewer #2: Yes

Reviewer #3: Yes

3. Have the authors made all data underlying the findings in their manuscript fully available?

Reviewer #1: Yes

Reviewer #2: No

Reviewer #3: Yes

4. Is the manuscript presented in an intelligible fashion and written in standard English?

Reviewer #1: No

Reviewer #2: Yes

Reviewer #3: Yes

5. Review Comments to the Author

Reviewer #1: The authors conducted a study focusing on genomic changes by investigating the features of thymic lymphoma (TL) with and without Calorie restriction (CR) after X-ray irradiation in mice. The result highlighted the characteristics of genomic alterations caused by CR. The findings in this paper can serve as a foundation for future efforts to reduce the risk of secondary cancers following radiotherapy. In my opinion, the experimental design and methodology described in this paper are sound. However, there is a need to address the comments below.

1) Please describe the treatment period for CR and non-CR in mice, as this information was not included in the paper.

2) In the description of the copy number in Fig. 2, what does the "a" at the end of "(Estimated to be LOH)a" mean? Also, is there an explanation of the TL numbers in black, blue, and red anywhere in the paper?

3) The 65Kcal pie chart in Fig. 3 adds up to 101%; please check if this is correct.

4) The p-value in the S7 Table has a different number of digits after the decimal point than it does in the main text of the study; this inconsistency should be resolved.

5) The expression of Ki-67-positive cells decreased. Please explain why you think it decreased.

Reviewer #2: This article describes genomic changes during development of thymic lymphoma (TL) which exhibits resistance against CR between calorie restriction (CR) status. This article is an interesting article and includes novel findings. There, however, are concerns about the article.

Major Comments

1. CR has been reported to decrease not only spontaneous but also chemical- and radiation-induced carcinogenesis, whereas some tumors including mouse lymphomas show CR resistance. CR should affect all cells in the host. Please explain the rationale that genomic changes in TLs cause resistance against the functions of CR which prevent the carcinogenesis. Were the genomic changes identified in other organs which exhibit CR sensitive?

2. We think these genomic changes in CR group were induced by CR because the genomic changes were identified compared to non-CR group. How do the genomic changes affect the prevention of suppression of TL development by CR? Were tumors suppressed in non-CR group? Again, we cannot understand the rationale how the genomic changes cause resistance against the functions of CR which prevent the carcinogenesis.

3. The authors showed several loci including chromosomes 4, 11, 12, 15, and 19. Nobody can know if other loci are associated with tumor development. Please show the all data.

4. In this study, the genomic status was shown but the authors did not show any functional assessments. Please describe this limitation.

Reviewer #3: The authors investigate the mechanism of resistance of radiation-induced thymic lymphoma to calorie restriction. The findings are that lymphoma, unlike other cancers utilizes a diverse carcinogenic pathway through the cell division-related pathways are effectively affected by calorie restriction.

The manuscript is well written. The experiments are conducted rigorously. The data presented strongly confers with the conclusion made.

However, the authors fail to show that radiation at single dose exposure of 3.8 Gy cause thymic lymphoma.

The evidence quoted through the cross reference of their earlier publication is not convincing.

It is not clear how many animals developed thymic lymphoma in the group studied, and how uniform the tumor burden is between each animal.

6. PLOS authors have the option to publish the peer review history of their article (what does this mean?). If published, this will include your full peer review and any attached files.

Reviewer #1: No

Reviewer #2: No

Reviewer #3: No

---

## [Author Response · Author response to Decision Letter 0]

28 Dec 2022

Dear Dr. Hari S. Misra,

Thank you for considering our recent manuscript “Calorie restriction alters the mechanisms of radiation-induced mouse thymic lymphomagenesis” and for allowing us to resubmit a revised version after considering the reviewers’ comments.

As per your request, we have responded to each reviewer’s comments and outlined our revisions below. In addition, two of our terms in the original manuscript were apparently difficult to understand, so we have made the following corrections.

Page 2, line 8: “(non-CR)”

Page 6, line 3: “(TLs)”

 Reviewer #1

Comment 1: Please describe the treatment period for CR and non-CR in mice, as this information was not included in the paper.

Response 1: Thank you for the comment. The period of CR was the entire breeding period from 7 weeks of age to the time of sacrifice, as indicated in the subsection of Materials and Methods (page 6, line 17) and Fig 1. Therefore, the period of CR differed for each mouse.

Comment 2: In the description of the copy number in Fig. 2, what does the "a" at the end of "(Estimated to be LOH)a" mean? Also, is there an explanation of the TL numbers in black, blue, and red anywhere in the paper?

Response 2: Thank you for the thoughtful comment. The explanation of "a" is given in the legend of Fig 2 (page 14, line 1). The sentence describes how we judged LOH at the Cdkn2a locus in TL7 and TL27. The TL numbers in red and blue are also explained in the legend of Fig 2 as “TL numbers in red indicate TLs with interstitial deletion and blue indicates TLs with copy-neutral LOH” (page 13, line 17). Still, the TLs in black were not explained, and thus we added the following explanation to the legend.

Page 13, line 19: “The TLs that did not fall into the aforementioned categories are shown in black.”

Comment 3: The 65Kcal pie chart in Fig. 3 adds up to 101%; please check if this is correct.

Response 3: Thank you for pointing this out. We have corrected the notation of Fig 3 so that the percentages in the pie chart sum to 100%, i.e., by rounding to the first decimal place.

Comment 4: The p-value in the S7 Table has a different number of digits after the decimal point than it does in the main text of the study; this inconsistency should be resolved.

Response 4: Thank you for pointing this out. We have corrected the number of decimal places for the p-values in S7 table to match that of p-values presented in the main text.

Comment 5: The expression of Ki-67-positive cells decreased. Please explain why you think it decreased.

Response 5: Thank you for the comment. In S6 Fig, the number of Ki-67+ cells in the thymus was estimated by multiplying the weight of each individual thymus by the percentage of Ki-67+ cells in a tissue section. As shown in S6 Fig C, the calculated weight of Ki-67+ cells in a thymus gradually decreased with age after 7 weeks of age in both groups. From 1 to 4 weeks after starting CR (8–11 weeks of age), the calculated weight in the 65 kcal group decreased drastically compared with the 95 kcal group. Furthermore, the decrease in the thymus weight in the 65 kcal group was greater than in the 95 kcal group, and the thymus weight was consistently less at all time points, with the exception of the data at 15 weeks of age, when thymic enlargement occurred in both experimental groups. These data suggest that fewer cells were proliferating in the 65 kcal group than in the 95 kcal group, supporting our interpretation.

Reviewer #2

Comment 1: CR has been reported to decrease not only spontaneous but also chemical- and radiation-induced carcinogenesis, whereas some tumors including mouse lymphomas show CR resistance. CR should affect all cells in the host. Please explain the rationale that genomic changes in TLs cause resistance against the functions of CR which prevent the carcinogenesis. Were the genomic changes identified in other organs which exhibit CR sensitive?

Response 1: Thank you for the important comment. We consider that several mechanisms pertain to the inactivation of tumor suppressor genes associated with thymic lymphomagenesis and that CR may alter the selection of the cell of origin for the lymphoma. In the present study, we could classify genetic alterations of Ikzf1, a causative gene of TL, as being attributable to two inactivating mechanisms, i.e., the “two-hit mechanism” and “dominant-negative mechanism”. During the development of TL, the “two-hit mechanism” is mediated by cell proliferation whereas the “dominant-negative mechanism” is independent of cell proliferation. In previous studies, CR indeed reduced cell proliferation in multiple organs of mice [Lok E et al. Cancer Lett. (1990), Wang TT et al. Cancer Lett. (1997)]; in addition, we partially confirmed that CR can reduce cell proliferation in the thymus by calculating the weight of Ki-67+ cells, as shown in S6 Fig C. Intriguingly, TLs in which Ikzf1 was inactivated by the dominant-negative mechanism were dominant in the CR group, whereas the two-hit mechanism was dominant in the non-CR group. Thus, our results suggest that CR-mediated suppression of cell proliferation decreases the chance of Ikzf1 inactivation by the two-hit mechanism. In other words, under CR, inactivation of Ikzf1 by the dominant-negative mechanism, which is unaffected by cell proliferation, emerged in place of the two-hit mechanism to cause TL, suggesting that the cancer-suppressive effect of CR was limited. For organs such as lung and liver, for which CR has a large suppressive effect on tumorigenesis, the number of samples required for the analysis could not be obtained because of the suppression of carcinogenesis by CR, making it impossible to analyze genomic alterations. Still, the data we acquired suggest that the suppression of genomic alterations through cell proliferation by CR may greatly contribute to the suppression of carcinogenesis in these organs.

Comment 2: We think these genomic changes in CR group were induced by CR because the genomic changes were identified compared to non-CR group. How do the genomic changes affect the prevention of suppression of TL development by CR? Were tumors suppressed in non-CR group? Again, we cannot understand the rationale how the genomic changes cause resistance against the functions of CR which prevent the carcinogenesis.

Response 2: Thank you for the comment. We address this comment in our response to your Comment 1 above.

Comment 3: The authors showed several loci including chromosomes 4, 11, 12, 15, and 19. Nobody can know if other loci are associated with tumor development. Please show the all data.

Response 3: Thank you for the comment. For data pertaining to copy number aberrations of all chromosomes analyzed by aCGH, we submit supporting information for reviewer #2 in addition to the raw data uploaded to the database. For chromosomes other than those mentioned in the text, several TLs had trisomy of chromosomes 1 and 14 and deletion of Notch1 on chromosome 2, but there were no significant differences in the frequency of these aberrations between the CR and non-CR groups. We added the following text in the revised manuscript and added Notch1 information to the S7 Table.

Page 16, line 10: “Trisomy of chromosomes 1 and 14, as well as deletion of Notch1 on chromosome 2, were identified in multiple TLs, but the frequency of these TLs was not significantly affected by CR.”

Comment 4: In this study, the genomic status was shown but the authors did not show any functional assessments. Please describe this limitation.

Response 4: Thank you for the thoughtful suggestion. Functional analysis of the genomic mutations detected in these causative genes was not carried out in this study because most of the genomic alterations we detected have been reported to be associated with TL development. We added the following text with additional references (Ref No. 34-40) to the Results.

Page 16, line 19: “Most of the genomic alterations detected in this study i.e., point mutations [16] and aberrant isoforms of Ikzf1 [32, 34, 35], accumulation of mutant Trp53 [36], deletion [20] and point mutations [31] in the phosphorylation site of Pten have been reported to be associated with lymphomagenesis. In addition, some of the genomic alterations detected in this study have already been shown to interfere with protein function [37-40].”

Reviewer #3

Comment: The manuscript is well written. The experiments are conducted rigorously. The data presented strongly confers with the conclusion made.

However, the authors fail to show that radiation at single dose exposure of 3.8 Gy cause thymic lymphoma.

The evidence quoted through the cross reference of their earlier publication is not convincing.

It is not clear how many animals developed thymic lymphoma in the group studied, and how uniform the tumor burden is between each animal.

Response: Thank you for the thoughtful comment. To show the incidence of TLs after exposure to 3.8 Gy of X-rays and the number of mice used in our previous studies, we have added the following explanation to the subsection of Materials and Methods.

Page 6, line 6: “The incidence of TLs in that previous study [14] was 20% (12/60) and 14.3% (9/63) in the non-CR and CR groups, respectively, after exposure to 3.8 Gy X-rays irradiation. No TL was observed in the non-irradiated group (95 kcal group; 0/60, 65 kcal group; 0/60). In the additional experiment, all mice were irradiated with 3.8 Gy X-rays, and the incidence of TL was 17.2% (17/99) and 14.1% (14/99) in the non-CR and CR groups, respectively; all experiments were”

---

## [Editor Report · Decision Letter 1]

2 Jan 2023

Calorie restriction alters the mechanisms of radiation-induced mouse thymic lymphomagenesis

PONE-D-22-29050R1

Dear Dr. Kakinuma,

We’re pleased to inform you that your manuscript has been judged scientifically suitable for publication and will be formally accepted for publication once it meets all outstanding technical requirements.

Kind regards,

Hari S. Misra

Academic Editor

PLOS ONE
---

## [Editor Report · Acceptance letter]

13 Jan 2023

PONE-D-22-29050R1 

Calorie restriction alters the mechanisms of radiation-induced mouse thymic lymphomagenesis 

Dear Dr. Kakinuma:

I'm pleased to inform you that your manuscript has been deemed suitable for publication in PLOS ONE. Congratulations! Your manuscript is now with our production department. 

Kind regards, 

on behalf of

Professor Hari S. Misra 

Academic Editor

PLOS ONE